# Metabolism of the Cyanogenic Glucosides in Developing Flax: Metabolic Analysis, and Expression Pattern of Genes

**DOI:** 10.3390/metabo10070288

**Published:** 2020-07-14

**Authors:** Magdalena Zuk, Katarzyna Pelc, Jakub Szperlik, Agnieszka Sawula, Jan Szopa

**Affiliations:** 1Faculty of Biotechnology, Wroclaw University, Przybyszewskiego 63/77, 51-148 Wrocław, Poland; katarzyna.pelc@uwr.edu.pl (K.P.); jakub.szperlik@uwr.edu.pl (J.S.); agnieszka.sawula2@uwr.edu.pl (A.S.); 2Linum Fundation, pl. Grunwaldzki 24A, 50-363 Wrocław, Poland; szopa@ibmb.uni.wroc.pl

**Keywords:** cyanogenic glucosides, valine N-monooxygenase, linamarase, hydroxynitrylase, cyanoalanine synthase, flax, linamarin, lotaustralin, linustatin, neolinustatin

## Abstract

Cyanogenic glucosides (CG), the monoglycosides linamarin and lotaustralin, as well as the diglucosides linustatin and neolinustatin, have been identified in flax. The roles of CG and hydrogen cyanide (HCN), specifically the product of their breakdown, differ and are understood only to a certain extent. HCN is toxic to aerobic organisms as a respiratory inhibitor and to enzymes containing heavy metals. On the other hand, CG and HCN are important factors in the plant defense system against herbivores, insects and pathogens. In this study, fluctuations in CG levels during flax growth and development (using UPLC) and the expression of genes encoding key enzymes for their metabolism (valine N-monooxygenase, linamarase, cyanoalanine nitrilase and cyanoalanine synthase) using RT-PCR were analyzed. Linola cultivar and transgenic plants characterized by increased levels of sulfur amino acids were analyzed. This enabled the demonstration of a significant relationship between the cyanide detoxification process and general metabolism. Cyanogenic glucosides are used as nitrogen-containing precursors for the synthesis of amino acids, proteins and amines. Therefore, they not only perform protective functions against herbivores but are general plant growth regulators, especially since changes in their level have been shown to be strongly correlated with significant stages of plant development.

## 1. Introduction

Generally, it can be stated that plant defense mechanisms consist of a number of antimicrobial compounds. They synthesize a wide spectrum of specialized metabolites involved in the protection against herbivores, pests and pathogens. For example, phenolic compounds act as feeding deterrents for herbivores but their role in resistance against fungi and bacteria is more dynamic than their role against insects. Other groups of specialized metabolites, such as cyanogenic glycosides (CG), are also toxic against some fungi and bacteria, and even viruses, but are more effective as feeding deterrents against a number of herbivores, including insects.

Cyanogenic glucosides are widely distributed in nature, being found in more than 2600 species [1,2]. The cyanogenic glycosides, with the R–CH–CN core of the molecule derived from aromatic or branched-chain amino acids, are defined as glycosides of α-hydroxynitriles. The natural products of flax, linamarin, linustatin, lotaustralin and neolinustatin are composed of an α-hydroxynitrile type aglycone and of a sugar moiety, mostly D-glucose (please see Figure 1). The precursors of the monoglucosides linamarin and lotaustralin synthesis are valine and isoleucine, respectively. The initial step is the N-hydroxylation of valine and isoleucine followed by the formation of 2-methyl-propanal oxime or 2-methylbutanal-oxime, respectively, and its dehydration to yield 2-methylpropionitrile or 2-methylbutylnitrile. The addition of oxygen forms acetone/butanon cyanohydrin, which is then glycosylated by UDPG-glucosyltransferase to form linamarin and lotaustralin [3]. Additional glycosylation leads to cyanogenic diglucosides (linustatin and neolinustatin). The degradation of cyanogenic glycosides starts with the removal of the sugar moiety by the action of specific β-glycosidases (e.g., linamarase). The resulting cyanohydrins are relatively unstable and decompose either spontaneously or enzymatically in a reaction catalyzed by α-hydroxynitrile lyase to an aldehyde or a ketone, and free hydrogen cyanide (HCN) [4]. It has been generally assumed that the linamarase activity is the rate-limiting step [1].

HCN is toxic to aerobic organisms as an inhibitor of respiration and heavy metal containing enzymes. β-cyanoalanine synthase plays a pivotal role in the detoxification of cyanide. It catalyzes the reaction of cysteine and cyanide to form hydrogen sulfide and β-cyanoalanine. The latter is subsequently converted to asparagine in a reaction catalyzed by β-cyanoalanine hydrolase.

The roles of CGs and CN^−^ vary and are understood only to some extent [5]. They are important factors of plant defense systems against herbivores, insects and pathogens [6]. Cyanoglycosides may also play an important role in primary metabolism and could function as nitrogen containing precursors for amino acid and protein synthesis during seedling development [7]. Moreover, free cyanide, including that released from the cyanogenic glycosides, may act as a signaling molecule [8]. Beneficial for protecting plants against infection, cyanogens can also be toxic to vertebrates at the same time.

Cyanogenic glucoside presents in linseed (*Linum usitatissimum)* is an example of compounds that participate in protection against infection but are toxic for mammalian consumers of plant products. There are some other contradictory effects of cyanide. For example, tissue disruption by herbivores brings CGs together with breakdown enzymes in the cytosol to release hydrogen cyanide, which is toxic to animals. Plants, however, are stationary for seed dispersal and thus reproduction needs to attract animals which play an important role in the process of pollination. Another example is that for health reasons, CG biosynthesis was reduced in transgenic cassava lines, which unexpectedly resulted in reduced plant growth [9]. Thus, a fine balance is expected between CG synthesis and degradation, which guarantees that the plant will maintain normal growth and development when protected from herbivores. High concentration of cyanide in soil or plant tissue would have negative impact on photosynthesis through disturbing electron transport in the thylakoids membrane. A number of specialized metabolites are also expected to maintain balance.

In the presented work, we provide a comprehensive analysis of the correlation between the metabolism of cyanogenic glycosides, their detoxification processes, and the general metabolism of plants during their life cycle. Only such a comprehensive approach to this topic allows us to fully show the role of these specialized metabolites for plants. It is not without significance that the tests were carried out on flax (*Linum usitatissimum)*—a plant with a wide range of health, nutritional and industrial applications.

In this study, overproduction of sulfur amino acids in transgenic flax was used as an approach to accelerate the incorporation of cyanide to specialized metabolites (amino acids) instead of volatilization. Accordingly, an increase in the amount of glutathione, arginine, lysine, diamines and alkaloids in transgenic flax was detected. It suggests that redirecting a part of cyanide to additional synthesis is beneficial for plant productivity. Although the CG content was reduced, transgene growth did not change, showed normal growth at normal yield and, more importantly, was more resistant to *Fusarium* than control plants [10]. Such analysis, enriched with a genetically modified plant, allows us to describe the relationship between the cyanide detoxification process and general metabolism.

## 2. Results

### 2.1. Assessment of Changes in Cyanogenic Glycoside Levels during Flax Growth

In order to ascertain the significance of cyanogen glucoside for flax physiology, we analyzed the compound metabolism during plant growth and development. The life cycle of the flax plant consists of several stages and is initiated by the germination period. In the seeds of flax, the cyanogenic diglucoside linustatin and neolinustatin are accumulated in the endosperm, the latter in excess. In dormant seeds, diglucosides (linustatin and neolinustatin) are the dominant form of cyanogenic glycosides (4.09 and 6.49 mg/g DW, respectively). Germination is characterized by the appearance of fully developed cotyledons. During germination the content of cyanogenic diglucosides dramatically decreases (see Figure 2D,E). After germination, the cyanogenic monoglucoside linamarin and lotaustralin are formed in the seedling (Figure 2B,C). The linamarin concentration was significantly higher (5.6 mg/g DW in 3 days after sowing (DAS)) compared to lotaustralin (2.1 mg/g DW). Since valine is also in excess (25% more compared with isoleucine), we suggest that higher availability of the core molecule may determinate the predominant content of linamarin. The vegetative step is initiated by slow development and is followed by a period of fast growth. The plant is able to elongate several centimeters per day during exponential growth (Figure 2A). During this phase in the young seedling a sharp decrease in diglucoside (below 3% of seed content), concomitant with a significant increase in monoglucoside content, has been detected (24-fold increase). At the linear sub-phase of the vegetative phase of plant growth, diglucoside was present at a minimal level. The next phases consist of flowering and a maturation with four sub-phases: bud formation, capsule development, seed development and seed maturation [11]. Flowering starts about 50 days after the germination step and lasts about 15 days. Bud formation, capsule and seed development start at the end of the flowering step and last about 30 days until seed maturation. Shortly before flowering starts, the second peak of monoglucoside content appears (15.4-fold increase in seed content). At the bud formation stage, the monoglucosides are significantly reduced (42% higher level than in seeds) and the process of storing cyanogenic glycosides in a more highly glycosylated, more stable form, begins.

Detailed data on the content of cyanogenic glycosides in plants at various stages of development are presented as Appendix A.

### 2.2. Identification of Key Enzymes for the Metabolism of Cyanogenic Glycosides

Six homologs of CYP79D1 cytochrome P450 (identical in about 59–63%) and two CYP71E7 homologs (52–59%) of genes derived from edible cassava (*Manihot esculenta*) were identified in the flax genome as possible candidates on the linamarin and lotaustralin biosynthesis pathway [12]. These enzymes should show NADP oxidoreductase activity. In addition, it has been shown that despite the fact that linustatin and linamarin are synthesized from valine and neolinustatin, and lotaustralin are formed from isoleucine, the synthesis of these compounds is under the control of one enzyme complex [13]. A gene encoding linamarase (LIN)—the enzyme responsible for converting diglucosides to cyanogenic monoglycosides—was identified in a similar way. In order to convert the nitrogen contained in cyanogenic glycosides into amino nitrogen, it is necessary to release and absorb the hydrogen cyanide. O-hydroxynitrile lyase and β-CAS are enzymes involved in the assimilation of cyanide to produce asparagine. Based on the above reports and the results of our own in silico analysis (Blast, Phytozome), the coding sequences for key enzymes of cyanogen glycoside metabolism in flax were selected. In silico analysis has been experimentally verified by checking the presence of transcripts (amplification of potential sequence coding for the expected genes in the PCR reaction and product sequencing) in genetic material isolated from Linola variety flax. The nucleotide sequences of the identified transcripts have been deposited at GenBank, respectively: NADPH-oxidoreductase (Val-monooxygenase) accession number MT739544; Linamarase (β-glucosidase) accession number MT739545 and β-cyanoalanine synthase accession number MT739546. One of the identified sequences has already been in GenBank—hydroxynitrile lyase, accession number AF024588.1.

The observed increase in the amount of cyanogenic monoglycosides during the germination process and rapid (vegetative) growth is probably a combination of synthesis processes catalyzed by NADPH oxidoreductase (valine monooxygenase) and linamarase-catalyzed deglycosylation (Figure 2F,G). As was mentioned above, shortly before flowering starts, the second peak of monoglucoside content appears. At the same time a significant increase in the gene expression of NADPH oxidoreductase was detected (four times higher compared to young seedlings (5 DAS) in the flowering stage (65 DAS)).

Fluctuations in the level of cyanogenic glycosides in the generative growth phase appear to be due to an increase in the activity of linamarase (degradation of diglucosides) and enzymes involved in cyanide assimilation: β-cyanoalanine synthase (CAS) (Figure 2H) and hydroxynitrilase (HNL) (Figure 2I), involved in cyanide assimilation to produce asparagine. Expression of the gene encoding linamarase is upregulated more than 7-fold in 65 DAS and up to 17-fold in 85 DAS (in comparison to 5 DAS). A dramatic increase in the activity of genes encoding HNL (65 DAS) and CAS (85 DAS) was also observed, suggesting the intensive relocation of nitrogen from cyanide to protein via asparagine. For more detailed data, see the Appendix A.

### 2.3. Assimilation of CN^−^ into Amino Acids and Diamines

It is suggested that the excess of cyanide ions formed as a result of the breakdown of cyanogenic glycosides can be detoxified with the participation of sulfur-containing metabolites, e.g., cysteine, and lead to the production of nitrogen metabolites such as amino acids and amines. The proposed metabolic transformations enabling the incorporation of cyanide ions into amino acids and further metabolites are shown in Figure 1.

To confirm this hypothesis, plants with significantly increased (eight times compared to control plants) cysteine levels (transgenic plant overexpressing the yeast gene Met25 (encoding O-acetylhomoserine- and O-acetylserine sulfhydrylase)) were used.

At most flax development stages, no significant quantitative changes in the level of cyanogenic glycosides were observed between the control and modified plants (Figure 2B–E). However, at some stages of development, i.e., intensive vegetative growth (3–10 DAS) and generative plant development—the formation and maturation of seed bundles (50–80 DAS)—statistically significant differences were noted between these types of plants. Observed changes in the metabolism of cyanogenic glycosides are undoubtedly associated with altered activity of genes encoding key enzymes for these processes (Figure 2F–I). In plants with increased cysteine content, a lower activity of the gene encoding linamarase (50–85 DAS) can be observed. Particularly noteworthy is the increase in β-cyanoalanine synthase (CAS) activity throughout the life of the plant. This observation supports the supposition of the higher intensity of CN^−^ detoxification processes in these plants.

An analysis of the same data (amino acid and amine metabolism: potential nitrogen acceptors derived from the detoxification of cyanide ions) obtained during the determination of the metabolic profile (GC-MS technique) of the 60-day Met25 plants compared to the unmodified plants was performed (Figure 3). A significant increase in the content of metabolites that may be products of the detoxification of cyanide ions was observed. The level of asparagine, the direct product of cyanide assimilation by cysteine, in a CAS-catalyzed reaction, doubled, resulting in a 4-fold increase in the level of aspartate, more than twice the level of threonine and more than three times the level of methionine in transgenic plants compared to the control. The doubled level, in comparison to the control plant, of 2-ketobutyrate, the intermediate of amino acid catabolism, was indicated. A 1.8-fold increase in lysine content, 1.6-fold in glutamate and even 15.5-fold in arginine were also observed. These amino acids in the decarboxylating reaction may produce primary diamines such as ornithine, putrescine and higher homologs such as spermidine and spermine (6.6-, 1.7-, 4.46-, 4.5-fold increase, respectively). The level of cadaverine, a decarboxylation product of lysine (in flax is for the first time reported) is dramatically higher (5-fold) in Met25 expressing plants. This compound occurs in very small amounts in flax and probably not throughout the life of the plant (it is not present in seeds and young seedlings)—which was probably the reason why it has not been identified yet.

It has been reported that the protein content changes during development [14], however, it remains similar for both analyzed genotypes. The total protein content increased during the generative stage of flax development, reaching a maximum (30% of body mass) on the 50th day after the start of flowering (100 DAS). At the same time, the level of cyanogenic glucosides (both mono- and diglucosides) were reduced to a minimum (Figure 4).

## 3. Discussion

Hydrogen cyanide (HCN) is a simple and diffusible molecule that is produced by the breakdown of cyanogenic glucosides and as a co-product of ethylene biosynthesis. The function of HCN is largely unknown. Generally it is regarded as a phytotoxic agent and plays a protective role against herbivores [15]. However, recently accumulated data indicate that, apart from being toxic, cyanide may regulate physiological processes, such as seed germination, nitrate assimilation, production of reactive oxygen species or plant responses to some environmental stimuli, such as drought stress and salinity stress [16,17,18] or plant resistance to fungal and viral infection [17]. The observed fluctuations in the level of cyanogenic glycosides strongly correlate with physiological changes and stages of plant development.

In the seeds of flax, the cyanogenic diglucoside linustatin and neolinustatin are accumulated in the endosperm, the latter in excess. The life cycle of the flax plant is initiated by the germination period. After germination, the cyanogenic monoglucoside linamarin and lotaustralin are formed in the seedling. Afterward, the life cycle of the flax plant consists of a 30- to 35-day vegetative step with two sub-steps: an (ca. 25 day) exponential growth and a (ca. 10 day) linear growth period, which generally agreed with published data [11]. The vegetative step is initiated by slow development and is followed by a period of fast growth. During this phase in the young seedling a sharp decrease in diglucoside, concomitant with a significant increase in monoglucoside content, was detected.

Assuming that the identified level of cyanogenic glycosides in the plant will be the result of the ratio of de novo synthesis and decomposition processes (mobilization of diglycosides to monoglycosides and assimilation of degradation products to amino acids), thorough examination of the expression of genes encoding enzymes that are key to these processes was performed.

The ratio of valine monooxygenase/linamarase transcripts, which can characterize de novo synthesis over decomposition, increases. In the next step of the vegetative phase, and precisely at the fast-growing step, expression of the linamarase gene increases, resulting in a dramatic decrease in diglucoside content. At the linear sub-phase of the vegetative phase of plant growth, diglucoside was at a minimal level.

Both β-cyanoalanine synthase (CAS) and nitrilase (HNL) genes, coding enzymes involved in cyanide assimilation to produce asparagine, were also observed to have high activity, thus suggesting the relocation of nitrogen from cyanide to protein via asparagine. Their increase may be explained by the fact that proteins serve as an energy source in the germination phase and the major sources of nitrogen and carbon during subsequent stages of the life cycle of flax [19,20]. It was found that at least 1305 genes were significantly induced by HCN in *Arabidopsis thaliana* [21]. Interestingly, among those genes several were involved in plant responses to stress and plant hormone signal transduction pathways. This indicates that HCN may participate in plant growth and development and stress responses. It also suggests that HCN may play a dual role in plants, depending on its concentration [20]. At high concentrations it may be effective against herbivores and may have a regulatory function at lower concentrations. Most HCN produced in plants is detoxified quickly by β-cyanoalanine synthase [22]. The remaining HCN, if it exists, at a lower level, probably at a non-toxic concentration, may act in the control of some metabolic processes. In this study, we have analyzed this suggestion by using a flax plant overproducing cysteine (Met25 expressed plants) [10]. Since cyanide is detoxified with a sulfur-containing compound, it could be expected that if cyanide is produced in greater amounts than the detoxification pathway can utilize, additional asparagine may be generated along with other nitrogen-containing compounds. Thus, HCN can be considered as a molecule directly affecting the genes of various metabolic pathways and interacting with other molecules in the regulation of gene expression.

Nitrogen re-assimilation to amino acids resulted in their higher quantity in the transgenic plant. Relevant to this, amino acids such as methionine, serine and threonine, catabolize via common intermediate, 2-ketobutyrate, generating energy (ATP or GTP) in the absence of oxidative phosphorylation [23]. It is important that during the rapid growth phase of flax, which requires more energy input, substrate phosphorylation avoids overstretching ATP glycolytic reserves.

In contrast to several other specialized metabolites, there is as yet no indication of the induction of cyanogenic glucoside biosynthesis upon plant infection. This suggests that the metabolic cost for maintaining cyanogenic glucoside content in a plant body is higher than for compounds like phenolics, which are synthesized and highly accumulated upon induction. How this extra cost is balanced is as yet not known. Perhaps enhanced survival of individual accumulating cyanogenic glucoside through protection against infection balanced this extra cost.

Our recent study [24] showed that apigenin and luteolin glycosides constitute the major group of flavonoids in flax. Alterations in their levels correlate with plant growth, peaking at the flowering phase. A significant correlation between flavonoid 3′-hydroxylase (F3′H) gene expression and the accumulation of luteolin glycosides has been found, indicating that flavone biosynthesis during flax growth and development is regulated by the temporal expression of this gene (please see Appendix A). The evidence suggests that the track of the flavonoid’s biosynthesis pattern might serve as a tool for the precise description of plant growth and development stages. The research results presented in this work show an inverse correlation between the content of flavonoids and cyanogenic glycosides at the stage of flax flowering.

Thus, the data strongly suggest the existence of the functional link between flavonoids and cyanogen glucoside in plant growth and development. However, nothing is known as yet about the molecular background of this potent link. Recent studies have shed some light on this issue. The reaction of cyanide with cystine-containing proteins result in the formation of an S-cyanylated Cys motif. S-cyanylation of oxidized cysteines in the form of disulfide bridges causes a change in the properties and function of the proteins. For example, most of the enzymes of metabolism are inhibited in the dark by the oxidation of active site cysteines and reactivated in the light by the action of ferredoxin/thioredoxin systems. Cyanide modification of the enzymes in their oxidized state resulted in the irreversible inactivation of their activity as shown with peptidyl-prolyl cis-trans isomerase [25]. The recent data provide evidence that several NADPH oxidoreductase genes are affected by cyanide [25]. Valine N-monooxygenase, the key gene of cyanogen glucoside synthesis, and flavonoid 3′-monooxygenase, the enzyme converting apigenin to luteolin are both NADPH oxidoreductase (heme–thiolate protein). It is thus speculated that cyanide, either by S-cyanylation or due to metal ions chelating, might affect both enzyme’s activity and regulate by itself its precursor synthesis and synthesis of flavone compounds. If so, the inverse relation of cyanogen glucoside and luteolin in flax during plant growth might occur, and this was the case. Thus, the potent link might occur through direct interaction of the CN anion with the heme–thiolate protein enzyme involved in cyanogen glucoside and flavonoid synthesis. Detailed genetic analysis and further physiological studies will improve our understanding of the link between both pathways.

## 4. Materials and Methods

### 4.1. Plant Material

Flax seeds (*Linum usitatissimum* L., Linola 947), control plants and tenth generation of the homozygous line of a transgenic plant overexpressing the yeast gene Met25 (encoding OAH-OAS sulfhydrylase (OAH-OASTL)) (transgenic plant generation was carefully described in [10]), were field cultivated in Wroclaw on a semi-technical scale.

The flax plants were harvested at six stages of flax plant development: sprout (3 and 7 DAS (days after sowing)), seedling (14 DAS), vegetative—stem extension stage (21 and 35 DAS), flowering period (49 and 56 DAS/1 and 7 DAF (days after start of flowering)), ripening (70 and 100 DAS/21 and 51 DAF) and dry maturity (120 DAS/71 DAF). All samples were collected in the early afternoon to eliminate the day period. All experiments (gene activity and metabolite content measurements) were performed in at least six independent biological replicates (plants from different areas of the field) and additionally for each biological sample three technical replicates.

### 4.2. Cyanogenic Glycosides (CG) Extraction

The flax tissue used was, in the case of 1–21 day-old plants, whole plants (with roots), while for older plants, the stems with seed capsules and flowers, but without roots, were used. Collected samples were lyophilized. The fully mature dry seeds were also used for analysis.

The lyophilized tissue (or seeds) was ground in a Retsch mill (1 min, 30 Hz) to a fine powder and 10 mg of the sample was extracted three times with 70% methanol using an ultrasonic bath (15 min). After centrifugation (5 min, 5000 rpm, RT), the supernatants from all steps of extraction were collected together, evaporated under a vacuum and finally resuspended in 1 mL of 70% methanol.

### 4.3. Separation and Identification Method for Cyanogenic Glycosides (CG)

Cyanogenic glycosides were analyzed on Waters UPLC MS (Ultra-Performance Liquid Chromatography Mass spectrometry) system (Waters, Milford, USA) with a UV-Vis diode array detector and Xevo QTof MS System mass spectrometer, using an Acquity UPLC column BEH C18, 2.1 × 100 mm × 1.7 μm. The mobile phase was composed of solvent A (Milli-Q water with 0.1% (*v*/*v*) HCOOH, pH 3) and solvent B (acetonitrile with 0.1% (*v*/*v*) HCOOH) in a gradient flow: 93% A/75% B at 0.5 min, 0.5–12 min gradient to 70% A/30% B; 13–20 min gradient to 0% A/100% B; 21 min gradient at 95% A/5% B with a 0.4 mL/min flow rate. The column was kept at 50 °C and the sample at 4 °C. Analysates were ionized by ESI (Electrospray Ionisation in negative polarity with an extended dynamic range and normal sensitivity. Low mass was set to 50 Da, high mass to 1000 Da and scan time to 0.2 s. Capillary voltage was set to 2.20 kV, sampling cone to 30 and extraction cone to 4.0. Temperatures were set to 80 °C for the source and 400 °C for desolvation. Gas flow was set to 20 (L/h) for the cone and 800 (L/h) for desolvation. The compounds were identified by comparison of their retention time and mass spectra to those of their commercially available standard (Sigma) and quantified through comparison to a concentration standard curve of their respective standards run with each batch of samples.

### 4.4. Extraction of Metabolites for GC-MS Analysis

100–150 mg of fresh flax tissue (60 DAS) was frozen in liquid nitrogen, and then was ground in a Retsch mill (1 min, 30 Hz) to a fine powder. To each sample, 1400 μL of 100% methanol and ribitol, as an internal standard, were added. Samples were shaken for 15 min at 70 °C, and then centrifuged at 18,000 rpm for 10 min. To the supernatant, 750 μL of chloroform and 1500 μL of water was added. Samples were then vortexed and centrifuged for 15 min at 2880 rpm. Finally, 150 μL of upper phase was speed-vacuum dried.

### 4.5. GC-MS Metabolite Profiling

To each sample, 40 μL of methoxyamine hydrochloride (20 mg/mL in pyridine) was added and the samples were shaken for 2 h at 37 °C, followed by derivatization with N-methyl-N-trimethylsilyltrifluoroacetamide and analyzed using GC–MS. The GC–MS system consisted of a GC 8000 gas chromatograph, an AS2000 auto sampler, and a Voyager quadrupole mass spectrometer (ThermoQuest, Manchester, UK). The chromatograms and mass spectra were evaluated using the MASSLAB program (ThermoQuest, Manchester, UK). The retention time and mass spectral library peak quantification of metabolite derivatives were implemented within the MASSLAB method format.

### 4.6. Gene Expression Analysis with Quantitative PCR

The level of mRNA for the analyzed genes was determined by real-time PCR. The analysis was carried out for at least three independent biological samples (whole plants were used for plants aged 1–21 days, and for older plants—obtained from field conditions—aerial parts, without seed capsules and flowers, were used). Complete RNA isolation, cDNA synthesis and real-time PCR were carried out as described earlier. Changes in the levels of analyzed transcripts are presented as relative quantification of the reference gene (actin). The sequences of the primers used to analyze gene expression are presented in the Appendix A.

### 4.7. Statistic Analysis 

One-way analysis of variance (ANOVA) was used to assess the effect of genetic modification on cyanogenic glycosides levels. Additionally, the two-way ANOVA analysis was performed and used to identify the simultaneous impact of the plant developmental stage and also carried out genetic modification, *p* value < 0.001 has been recognized as statistically significant.

## 5. Conclusion

Based on these observations, the following pathway for the in vivo mobilization and metabolism of cyanogenic glucosides in flax is proposed: storage of diglucosides (in seeds); deglucosylation and monoglucoside accumulation in seedling cleavage by linamarase at an early step of the vegetative phase; reassimilation of HCN to non-cyanogenic compounds at an exponential step of the vegetative phase; de novo synthesis of monoglucoside at an initial step of the flowering phase; reassimilation of −CN to amino acids, glutathione, diamines and alkaloid at bud formation; capsule and seed development steps of the maturation phase; part of cyanogenic monoglucoside conversion to diglucoside at the seed maturity step of the maturation phase. Nitrogen re-assimilation to amino acids resulted in their higher quantity in the transgenic plant. Relevant to this, amino acids such as methionine, serine and threonine, catabolize via common intermediate, 2-ketobutyrate, generating energy (ATP or GTP) in the absence of oxidative phosphorylation [23]. It is important that during the rapid growth phase of flax, which requires more energy input, substrate phosphorylation avoids overstretching ATP glycolytic reserves.

In summary, to our best knowledge we demonstrate for the first time the “metabolic pressure” of the excess of cysteine availability in redirecting cyanide to other pathways than amino acids and thus protein biosynthesis. We have found that CN anion can be re-assimilated to basic amino acids, such as arginine, lysine and ornithine and thus to polyamines such as putrescine, spermidine, spermine and cadaverine. Cadaverine, a decarboxylation product of lysine in flax, is for the first time reported. Finally, for the first time we demonstrate the link between cyanogen and flavonoid pathways.

## Figures and Tables

**Figure 1 metabolites-10-00288-f001:**
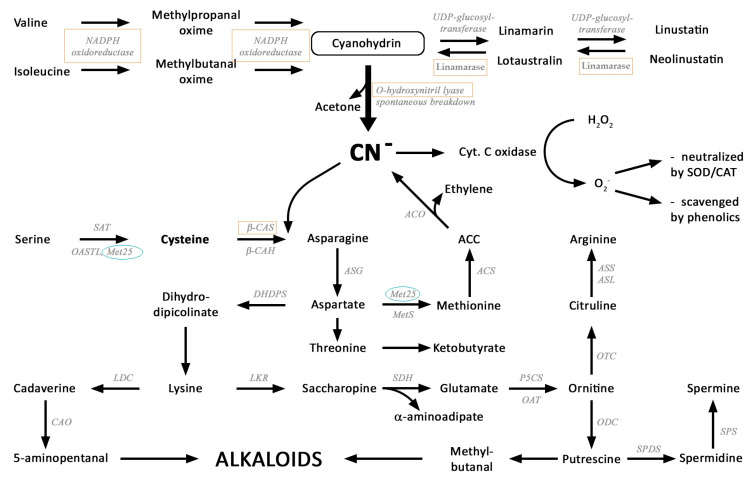
Metabolism of cyanogenic glucosides. Key enzymes marked on the scheme: biosynthesis: NADPH oxidoreductase (Val-monooxygenase), UDP-glucosyl-transferase; catabolism: linamarase (β-D-glucosidase), O-hydroxynitryl lyase; CN^−^ ions asymilation: β-CAS -β-cyanoalanine synthase, β-CAH-β-cyanoalanine hydratase, SAT-serine O-transacetylase, OASTL-O-acetylserine (thiol) lyase, ASG-asparaginase, MetS-methionine synthase, ACS-ACC synthase, ACO-ACC oxidase, DHDPS-dihydro-dipicolinate synthase, LDC-L-lysine decarboxylase, CAO-cadaverine oxydase, LKR-lysine-ketoglutarate reductase, SDH-sacchropine dehydrogenase, P5CS-pyrroline-5-carboxylate synthase, OAT-ornithine aminotransferase, ODC-ornithine decarboxylase, SPDS-spermidine synthase, SPS-spermine synthase, OTC-ornithine transcarbamylase, ASS-argininosuccinate synthetase, ASL-argininosuccinate lyase; ROS scavenging processes: CAT-catalase, SOD-superoxide dismutase. Exogenous gene Met25-O-acetylhomoserine-O-acetylserine (OAH-OAS) sulfhydrylase from *Saccharomyces. cerevisiae*. Genes whose expressions have been studied and described in this work are marked with red boxes. The blue ellipse marked the activity site of the Met25 yeast gene introduced into transgenic plants.

**Figure 2 metabolites-10-00288-f002:**
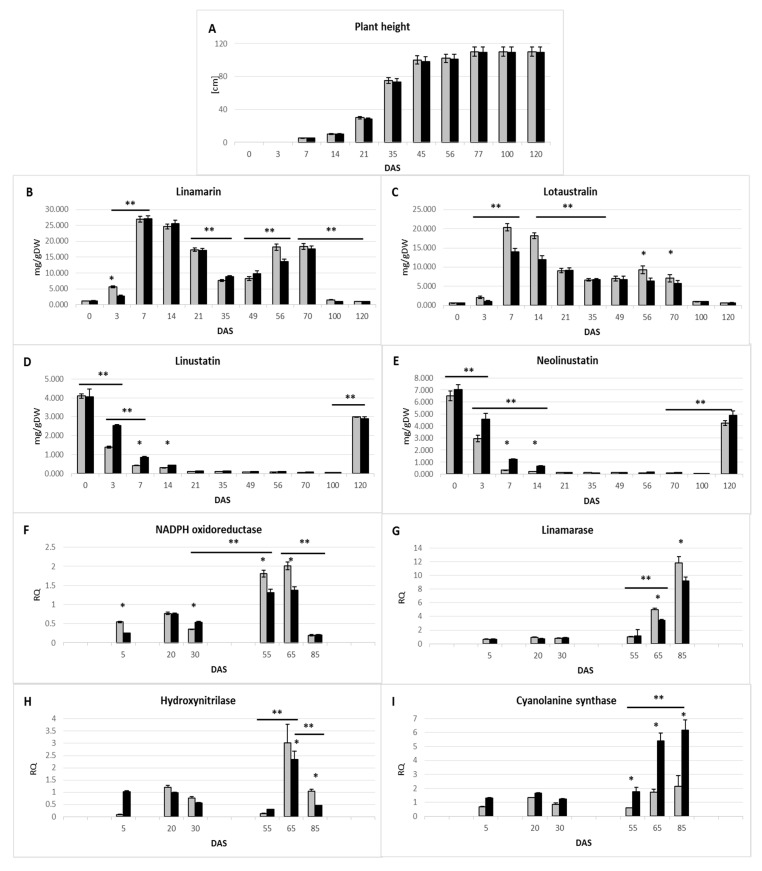
Fluctuation in cyanogenic glucoside levels and expression of GC metabolism genes during plant growth in control and Met25 expressing plants (comparison). Plant growth in height (**A**); fluctuation of cyanogenic monoglucoside level (**B**,**C**); fluctuation of cyanogenic diglucoside level (**D**,**E**); the mRNA level of GC metabolism genes: NADPH oxidoreductase (**F**), linamarase (**G**), β-cyanoalanine synthase (**H**) and hydroxynitrilase (**I**) obtained from the real-time RT-PCR analysis. DAS, days after sowing; RQ, relative quantity. Data represent the mean ± standard deviations from at least three independent experiments. Data constitute the mean value ± SD from at least three technical repeats of three independent biological samples. The significance of the differences between the transgenic plant and control was determined by Student’s t-test. Asterisk indicates *p* < 0.05. For an indication of developmental changes, the ANOVA test was used—*p* value = 0.001 was indicated by asterisks and the line indicates the compared data range.

**Figure 3 metabolites-10-00288-f003:**
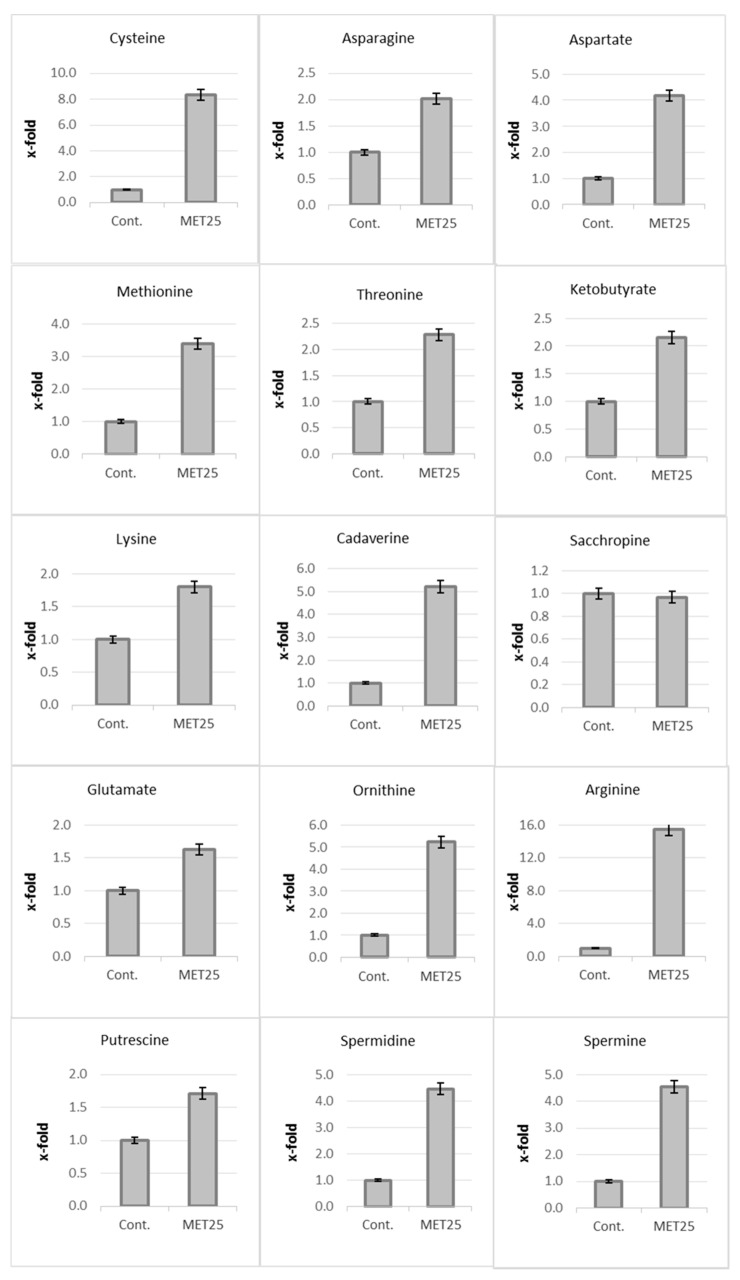
Content of metabolites resulting from detoxification and assimilation of cyanide ions. Met25 plants compared to control plants (Cont.); six independent measurements ± SD are presented.

**Figure 4 metabolites-10-00288-f004:**
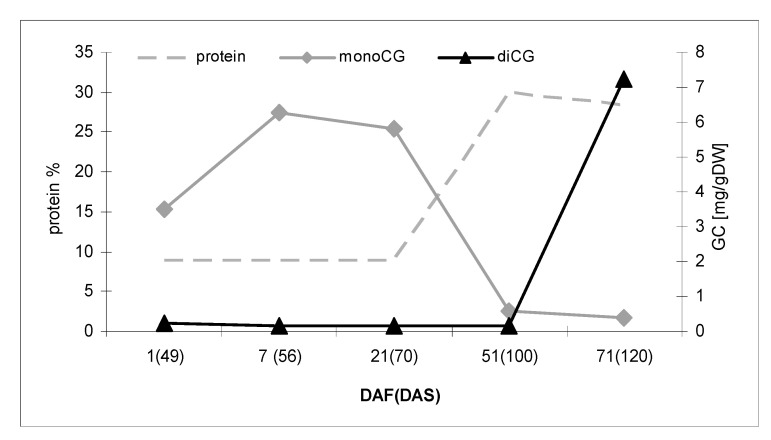
Schematic representation of changes in total protein content (% of DW) and cyanogenic mono- and diglucoside levels during generative development of flax (control) plant. DAF, days after flowering; DAS, days after sowing.

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
