# Peer review of "Metabolism of the Cyanogenic Glucosides in Developing Flax: Metabolic Analysis, and Expression Pattern of Genes"

_metabolites, 2020, doi:10.3390/metabo10070288_

Round 1

Reviewer 1 Report

The article by Zuc et al. concerns with the changes in cyanogenic glucosides in flax during the plant growth. The article is well written and within the journal scope, but there are some aspects that could be improved to better convey the information to the readers.

- The authors should better clarify in the introduction the novelty of the work as well as the objective: while it is possible to infer that it is to study the relationship between the cyanide detoxification process and general metabolism, the novelty of the approach is not emphasized.

- In Figure 1, as well as in Tables S1 and S2, there is a lack of comparison between the native flax and Met25 and, while the authors tell us the results of ANOVA analysis in the text, no such information is reported in the figure. As it is the figure and the tables are interesting to observe the trend of the variables during the plant development, but another figure should be used comparing the variables (i.e. CG content) at the same DAS for the two varieties. Figure 1 could be moved as it is in supporting information.

- In the section 2.3 it is not clear if the molecules reported in Figure 2 are all the examined ones, or only the variables significant to ANOVA analysis. If so, that should be added to figure caption. Moreover, in that figure report the absolute value and not a relative one.

- Is it possible for the authors to carry out a multivariate PCA analysis on UPLC data? That could provide further insight on the metabolic processes.

Some minor points:

- The authors should use accepted name of flax the first time it is introduced in the manuscript, and not only in 4.1 section

- at lines 154-155 the comma should not be after “lead”, but after “cysteine”

- Figure 3 refers to Control or Met25 plants? Please specify in the caption

Author Response

The manuscript has been significantly changed as suggested by the editor and reviewers.

Below is a list of the most important changes:

  1. The manuscript title has been slightly changed as suggested by the reviewer. All changes in the manuscript have been highlighted using the "track changes" option
  2. The completely new figure was added (Fig 1) depicting the metabolic transformations of cyanogenic glycosides (their synthesis, catabolism and assimilation of cyanide ions to amino acids and biogenic amines). The diagram also shows the relationship of the metabolic pathway in question with other important processes in the plant organism such as ROS scavening or flavonoid metabolism.

[Proposed placement of a new figure line 55]

Inserting a new figure changed the numbering of subsequent ones.

  1. After reviewing the suggestions of reviewers, it was decided to remodel Figure 2 (previously Fig1). Data on control and transgenic plants were placed on one diagram, which facilitates comparison of these two types of plants, while not adversely affecting the ability to track changes in metabolite levels / gene expression in development. This change made the separation of some metabolites - mono- and diglycosides into separate diagrams. We believe that this change will significantly improve the transparency of the content and facilitate their perception. Obviously, the description in the text and the figure caption accordingly has been changed. In addition, statistical markers have been placed on the diagrams to identify statistically significant changes.
  2. The description of identification of genes (coding sequences) was, according to the reviewer's suggestion, placed in the results description section 2.2 (moved from discussion )- line 178-190

5.The name Val-monooxygenase has been changed to NADPH oxidoreductase - which in our opinion better reflects the function of this - enzyme (both names function in parallel in the scientific literature.) Change carried out throughout the manuscript test and figures.

6.Additional metabolites have been added to the data presented in Figure 3 (previously 2), which in our opinion enriches the message. The fragment about linking metabolism of cyanogenic glycosides with flavonoids during plant development was added to the discussion.(line 339-371 ) An appropriate figure has been added to the supplementary materials (Supp Fig1).

  1. The description of the statistical analysis has been moved to the Materials and Methods section.
  2. The description of the method (section 4.6) has been rephrased.

Rev 1 answer:

  1. The article's introduction has been enriched with content highlighting novelty and originality.
  2. The original figure 1 - and now Figure 2 -is completely rebuilt in order to avoid the weaknesses identified. The presentation of data on control plants modified in one plot makes it easier to compare them with each other - which was indeed difficult in the previous version of the figure. Additional information resulting from the analysis ANOVA has  been shown in the new version of the figure.

Tables S1 and S2 are left in their original version - they are only a clarification (numerical) of the data presented in the Figure for a more discerning reader.

  1. Figure 3 (formerly 2) presents data on the metabolism of amino acids and amines - potential nitrogen acceptors resulting from the detoxification of cyanide ions - this is, of course, part of the data obtained during the analysis of the metabolic profile of the plants. The group of presented data has been selected for the needs of this statement / this publication.

An appropriate clarifying sentence was added in the manuscript text - lines 236-237

The publication uses relative values to better illustrate the differences between the two types of plants analyzed. additionally, for some of the metabolites analyzed, we only have relative data - derived from the analysis of the total metabolic profile of plants - this form of data presentation allowed us not to give up some of the relevant data - and did not deplete, in our view, the value of the publication. In this part of the work, the data was used only to illustrate the processes of nitrogen incorporation / assimilation from cyanides in the analyzed plant types and the metabolic effect that can be achieved by influencing the intermediates of this process.

  1. An additional PCA analysis of data obtained from UPLC and metabolic profile analyzes was performed - the results of these analyzes confirmed previous conclusions - the most important factor generating variability / differentiating data/ are development stages. an additional differentiating factor is, of course, nitrogen metabolism (amino acids, amines) - these differences result from the genetic modification carried out. Because these conclusions are consistent with earlier - obtained from two-way ANOVA analysis - the manuscript was not supplemented with them.
  2. Thank you very much for additional comments and as a result:

- the flax species name has been added in the introduction

- comma positions on line 154-155 have been changed (currently 211-212).

- in the description of figure 3 (now 4) information has been added that it describes the control plants. Data for modified plants are very similar - that's why they were abandoned so as not to complicate the image.

Reviewer 2 Report

In this manuscript untitled “Metabolism of the cyanogenic glucosides in developing flax: biochemical characterization, and expression pattern of genes”, Zuk et al. report the analysis of cyanogenic glycoside contents and of the expression of genes involved in the metabolism of these molecules along the development of flax.

I think this manuscript needs major improvements before it can be published in Metabolites.

Major concern:

Authors analyzed the contents of molecules during development from germination to flowering and seed formation. I think a more detailed analysis of CGs should be done. In fact, it could be expected that the accumulation of CGs is more important in young organs to protect them form predators. Analysis of dissected flower, young leaves, old leaves, roots would be better to gain more information regarding physiology or biology. In fact, it is known that these compounds are usually accumulated at higher levels in young organs for example or in flowers. Moreover authors wrote in the Materials and methods part that they collected the whole plants at the beginning of the experiments and then they collected only the aerial parts. To me, this could introduce mistakes. In addition, PCR experiments were only done with RNA extracted from stems. It makes difficult the comparison of the expression data and those from metabolite analysis.

In the introduction (line 42), the synthesis of linamarin from valine is described. What about the synthesis of lotaustralin and the two other molecules? I think a figure should be added to describe the metabolism of cyanogenic glycosides (including catabolism). It will be useful for non-specialists of this pathway and to understand the text between line 40 and line 50. It will also be helpful to indicate the genes that were chosen for qRT-PCR.

Line 73-74: to me the cyanide is not converted in specialized metabolites but incorporate to form amino acid.

Line 77-79: the resistance to fusarium was not investigated in this paper. I think this part should be placed in the discussion as the lines 80-85.

Line 93: insert a reference to Fig. 1C (contents of linustatin and neolinustatin).

Line 98-99: I am not sure that substrate availability explains the contents of the 2 CGs. To address this point, authors should at least measure gene expression for example.

The SD are not shown in Fig1 A, B, C, F, G and H.

Legend of Figure 1: it is not a schematic representation. Authors did not measure gene activity but gene expression.

In the section 2.2, authors should mention why they chose these genes and how they did to find them (blast…). I did not find the gene accession numbers. They should be added.

The y axis should be modified. In fact, the expression of valine monooxygenase is difficult to see and we can barely see the differences in the level of gene expression between the different developmental stages.

Again these genes should be placed on a biochemical pathway because it is difficult to know if the chosen genes are involved in catabolism or synthesis of the CGs.

Line 157: what is OAH-OAS sulfhydrylase?

Comparison of linola variety and Met25 transgenic plants is very difficult to follow. Authors mentioned differences in the amounts of CGs but based on the figure it is not really obvious. This part should be improved. Data from WT and transgenic plants could be placed on the same figure.

When measuring the protein content, I would suggest to add the analysis of the transgenic plants.

Line 208-209: I do not understand why the data presented in the manuscript support the hypothesis mentioned from line 201 to line 207.

Line 210-223: this should be moved to the result part. A more detailed description of the genes is required as mentioned above.

Minor points:

The title should be modified. I do not think the authors performed biochemical characterization but metabolic analysis.

Line 15: (UPLC) should be replaced by “using UPLC”

Line 18: cultivar instead of cultivars because only one cultivar was analyzed.

Line 30: involved instead of specializing

The words “secondary metabolites” should be replaced by “specialized metabolites” throughout the manuscript.

Author Response

The manuscript has been significantly changed as suggested by the editor and reviewers.

Below is a list of the most important changes:

  1. The manuscript title has been slightly changed as suggested by the reviewer. All changes in the manuscript have been highlighted using the "track changes" option

  1. The completely new figure was added (Fig 1) depicting the metabolic transformations of cyanogenic glycosides (their synthesis, catabolism and assimilation of cyanide ions to amino acids and biogenic amines). The diagram also shows the relationship of the metabolic pathway in question with other important processes in the plant organism such as ROS scavening or flavonoid metabolism.

[Proposed placement of a new figure line 55]

Inserting a new figure changed the numbering of subsequent ones.

  1. After reviewing the suggestions of reviewers, it was decided to remodel Figure 2 (previously Fig1). Data on control and transgenic plants were placed on one diagram, which facilitates comparison of these two types of plants, while not adversely affecting the ability to track changes in metabolite levels / gene expression in development. This change made the separation of some metabolites - mono- and diglycosides into separate diagrams. We believe that this change will significantly improve the transparency of the content and facilitate their perception. Obviously, the description in the text and the figure caption accordingly has been changed. In addition, statistical markers have been placed on the diagrams to identify statistically significant changes.
  2. The description of identification of genes (coding sequences) was, according to the reviewer's suggestion, placed in the results description section 2.2 (moved from discussion )- line 178-190

5.The name Val-monooxygenase has been changed to NADPH oxidoreductase - which in our opinion better reflects the function of this - enzyme (both names function in parallel in the scientific literature.) Change carried out throughout the manuscript test and figures.

6.Additional metabolites have been added to the data presented in Figure 3 (previously 2), which in our opinion enriches the message. The fragment about linking metabolism of cyanogenic glycosides with flavonoids during plant development was added to the discussion.(line 339-371 ) An appropriate figure has been added to the supplementary materials (Supp Fig1).

  1. The description of the statistical analysis has been moved to the Materials and Methods section.
  2. The description of the method (section 4.6) has been rephrased.

    Rev 2 answers:

    1. I agree with the reviewer that it would be interesting to analyze the level of cyanogenic glycosides in individual plant organs: roots, young leaves, older tissues, flower buds, immature seminal capsules, side shoots, etc. We are currently at the stage of such analysis. However, for the needs of this work, we decided to analyze the whole stems (ground part). The extraction of nucleic acids and glycosides from immature and mature seminal capsules must be carried out according to other procedures due to the high content of lipids and tannins and it cannot be performed simultaneously with extraction from other parts of the stem. In the analyzes of more mature tissues presented, the root analysis was also abandoned - the material was obtained from field conditions - where flax is sown relatively densely (to avoid lodging) and obtaining a root from one plant results in the destruction of the neighbors' root structure.

    In addition, linen produces very small narrow leaves adhering to the stem - which is why they were analyzed together with the stem. During the growth and aging of the tissues, these leaves dry up and often fall off.

    Flowers of flax appear in the early morning and petals fall off in the afternoon - it is difficult to obtain sufficient material for research. All your difficulties influenced our decision regarding the selection of material for research.

    1. The introduction of the publication was supplemented with a description of the synthesis of all cyanogenic glycosides (lines 42-54). in addition, there is a suitable figure that will certainly facilitate the understanding of the course of the described metabolic changes. The analyzed genes (qRT-PCR) are marked in the diagram.
    2. The introduction of the publication was supplemented with a description of the synthesis of all cyanogenic glycosides (lines 42-54). in addition, there is a suitable figure that will certainly facilitate the understanding of the course of the described metabolic changes. The analyzed genes (qRT-PCR) are marked in the diagram.
    3. Indeed, cyanides are not converted but incorporated into amino acids - which has been corrected in the manuscript (line 104).
    4. A reference to the correct figure has been added – line 127, 128.
    5. Of course, the claim that substrate availability is a factor that influences the amount of individual cyanogenic glycosides is only a hypothesis. However, knowing, as it is reported in the literature, that both the strips are probably carried out by the same enzyme, one can draw such a conclusion. moreover, such speculation also appears in other publications describing the metabolism / biosynthesis of cyanogenic glycosides.
    6. Fig 1 (currently 2) in its latest form contains SD.
    7. Legend of Fig 1 was improved according to Rev. suggestion.
    8. The process of identifying the coding sequences for the analyzed genes has now been described quite accurately and is included in section 2.2. Our genes do not have accession numbers. They were found on the basis of homologies to known genes of metabolism of cyanogenic glycosides and verified by sequencing of fragments obtained as a result of PCR (performed on flax mRNA).
    9. In the present form of Figure 2, the graphs showing the expression of individual genes have been separated - which allowed the use of more suitable scale of the obtained individual results. In our opinion, the current charts are much clearer and the problem regarding the difficulty in reading the values for individual genes has been solved.
    10. O-acetylhomoserine- O-acetylserine sulfhydrylase = OAH-OAS sulfhydrylase – please see line 216-217.

    12 .Thank you very much for the suggestion of placing control data and modified plants on one figure - a very good solution, it would seem obvious but sometimes the simplest remains hidden :). As you suggested, a new version of the figure has been prepared - Fig 2.

    1. The values describing the protein level for transgenic and control plants are very similar - we do not observe significant quantitative changes, only qualitative (change in the amino acid composition - specifically the proportion of sulfur amino acids - Cys, Met etc. as shown in Figure 3). In order not to complicate the image, the data are presented only for control plants.
    2. The location of this sentence is unfortunate, they have been removed. Further in the discussion hypotheses that can be drawn and confirmed based on the results presented here are presented.
    3. The gene identification description was moved as suggested to the results section.

    16.The title of the publication has been changed as suggested.

    line 15- UPLC changed to using UPLC

    line 18- one cultivar - of course :)

    line 30 - corrected - involved was used.

    Throughout the article, secondary metabolites have been replaced with specialized metabolites in accordance with the suggestion.

Round 2

Reviewer 1 Report

The authors substantially improved the article, and it can be published as it is

Author Response

Thank you very much for your valuable comments and time.

Reviewer 2 Report

The authors deeply improved the overall quality of their manuscript. They appropriately respond to all my comments.

The main issue is the sequences of the genes which are not included in the manuscript: they should be deposited on genebank and the accession provided in the manuscript.

I have few minor comments:

  • The 2 sentences line 37-39 must be combined. The core of the molecule is described twice.
  • A reference to Fig. 1 should be introduce line 42
  • I would suggest to remove the sentence line 57-58: metabolic processes -synthesis....
  • Legend of Fig. 1: Line 60-61 could be replaced by "Metabolism of cyanogenic glycosides". Line 74-75: the sentence should be improved (Red boxes...).
  • Line 95 and other parts of the introduction: authors mention that CN is toxic for animals but it is also toxic for the plant itself. 
  • Line 102: In the present work, we provide a comprehensive analysis of the.... during their life cycle.
  • Line 223-224: "potential sequences encoding genes..". I think this sentence should be removed.
  • Line 227: this sentence should be improved.
  • Line 250-255: this section should be improved (from although to seed bundles).
  • For the protein level, authors should add a sentence to indicate that the level of proteins in the two genotypes are similar.
  • line 205: of indepent three biological samples" should be replaced by "three independent biological samples".

Author Response

Thank you very much for valuable comments leading to the improvement of the quality of our manuscript.

We agree with the reviewer that the right step is to place the sequence of our genes in the Genebank and we are currently preparing for this task (digesting recent experiments). Unfortunately, the current pandemic situation has slowed down our work in this area (significant limitations of work opportunities in the laboratory), while we plan to fill this gap in the future.

  • Sentences line 37-39 were combined. Thank you very much for your correction.
  • The reference to Fig 1 were added on line 42.
  • Sentence in line 57-58 were removed.
  • The legend of Fig 1 was improved as suggested.
  • The suggestion regarding the potential negative impact of cyanide ions on plant metabolism has been added to the introduction – line 93-95.
  • The sentence presented in line 102 was also corrected.
  • The mentioned sentence (line223) was removed.
  • Line 227 - the sentence has been reworded to make it easier to understand.
  • Mentioned section (line 250-255) has been improved and a new version introduced.
  • Information on similar levels of total protein in both genotypes has been included in the text.

This manuscript is a resubmission of an earlier submission. The following is a list of the peer review reports and author responses from that submission.